# Manifold Modeling in Embedded Space: A Perspective for Interpreting "Deep Image Prior"

## Abstract

Deep image prior (DIP) (Ulyanov et al., 2018), which utilizes a deep convolutional network (ConvNet) structure itself as an image prior, has attracted huge attentions in computer vision community. It empirically shows the effectiveness of ConvNet structure for various image restoration applications. However, why the DIP works so well is still unknown, and why convolution operation is useful for image reconstruction or enhancement is not very clear. In this study, we tackle these questions. The proposed approach is dividing the convolution into "delay-embedding" and "transformation (*i.e.,* encoder-decoder)", and proposing a simple, but essential, image/tensor modeling method which is closely related to dynamical systems and self-similarity. The proposed method named as manifold modeling in embedded space (MMES) is implemented by using a novel denoising-auto-encoder in combination with multi-way delay-embedding transform. In spite of its simplicity, the image/tensor completion, super-resolution, and deconvolution results of MMES are quite similar even competitive to DIP in our extensive experiments, and these results would help us for reinterpreting/characterizing the DIP from a perspective of "low-dimensional patch-manifold prior".

## 1 Introduction

The most important piece of information for image/tensor restoration would be the "prior" which usually converts the optimization problems from ill-posed to well-posed, and/or gives some robustness for specific noises and outliers. Many priors were studied in computer science problems such as low-rank representation (Pearson, 1901; Hotelling, 1933; Hitchcock, 1927; Tucker, 1966), smoothness (Grimson, 1981; Poggio et al., 1985; Li, 1994), sparseness (Tibshirani, 1996), non-negativity (Lee & Seung, 1999; Cichocki et al., 2009), statistical independence (Hyvarinen et al., 2004), and so on. Particularly in today's computer vision problems, total variation (TV) (Guichard & Malgouyres, 1998; Vogel & Oman, 1998), low-rank representation (Liu et al., 2013; Ji et al., 2010; Zhao et al., 2015; Wang et al., 2017), and non-local similarity (Buades et al., 2005; Dabov et al., 2007) priors are often used for image modeling. These priors can be obtained by analyzing basic properties of natural images, and categorized as "unsupervised image modeling".

By contrast, the deep image prior (DIP) (Ulyanov et al., 2018) has been come from a part of "supervised" or "data-driven" image modeling framework (*i.e.,* deep learning) although the DIP itself is one of the state-of-the-art unsupervised image restoration methods. The method of DIP can be simply explained to only optimize an *untrained* (*i.e.,* randomly initialized) fully convolutional generator network (ConvNet) for minimizing squares loss between its generated image and an observed image (*e.g.,* noisy image), and stop the optimization before the overfitting. Ulyanov et al. (2018) explained the reason why a high-capacity ConvNet can be used as a prior by the following statement: *Network resists "bad" solutions and descends much more quickly towards naturally-looking images*, and its phenomenon of "impedance of ConvNet" was confirmed by toy experiments. However, most researchers could not be fully convinced from only above explanation because it is just a part of whole. One of the essential questions is why is it ConvNet? or in more practical perspective, to explain what is "priors in DIP" with simple and clear words (like smoothness, sparseness, low-rank etc) is very important.

In this study, we tackle the question why ConvNet is essential as an image prior, and try to translate the "deep image prior" with words. For this purpose, we divide the convolution operation into

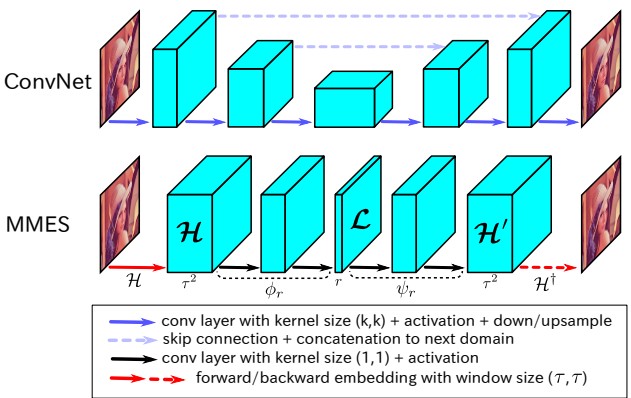

Figure 1: Comparison of typical auto-encoder ConvNet and the proposed MMES network.

"embedding" and "transformation" (see Fig. 9 in Appendix). Here, the "embedding" stands for delay/shift-embedding (*i.e.,* Hankelization) which is a copy/duplication operation of image-patches by sliding window of patch size $(\tau, \tau)$. The embedding/Hankelization is a preprocessing to capture the delay/shift-invariant feature (*e.g.,* non-local similarity) of signals/images. This "transformation" is basically linear transformation in a simple convolution operation, and it also indicates some non-linear transformation from the ConvNet perspective.

To simplify the complicated "encoder-decoder" structure of ConvNet used in DIP, we consider the following network structure: Embedding $\mathcal{H}$ (linear), encoding $\phi_r$ (non-linear), decoding $\psi_r$ (non-linear), and backward embedding $\mathcal{H}^\dagger$ (linear) (see Fig. 1). Note that its encoder-decoder part $(\phi_r, \psi_r)$ is just a simple multi-layer perceptron along the filter domain (*i.e.,* manifold learning), and it is sandwitched between forward and backward embedding $(\mathcal{H}, \mathcal{H}^\dagger)$. Hence, the proposed network can be characterized by Manifold Modeling in Embedded Space (MMES). The proposed MMES is designed as simple as possible while keeping a essential ConvNet structure. Some parameters $\tau$ and $r$ in MMES are corresponded with a kernel size and a filter size in ConvNet.

When we set the horizontal dimension of hidden tensor $\mathcal{L}$ with $r$, each $\tau^2$-dimensional fiber in $\mathcal{H}$, which is a vectorization of each $(\tau, \tau)$-patch of an input image, is encoded into $r$-dimensional space. Note that the volume of hidden tensor $\mathcal{L}$ looks to be larger than that of input/output image, but representation ability of $\mathcal{L}$ is much lower than input/output image space since the first/last tensor $(\mathcal{H}, \mathcal{H}')$ must have Hankel structure (*i.e.,* its representation ability is equivalent to image) and the hidden tensor $\mathcal{L}$ is reduced to lower dimensions from $\mathcal{H}$. Here, we assume $r < \tau^2$, and its low-dimensionality indicates the existence of similar $(\tau, \tau)$-patches (*i.e.,* self-similarity) in the image, and it would provide some "impedance" which passes self-similar patches and resist/ignore others. Each fiber of Hidden tensor $\mathcal{L}$ represents a coordinate on the patch-manifold of image.

It should be noted that the MMES network is a special case of deep neural networks. In fact, the proposed MMES can be considered as a new kind of auto-encoder (AE) in which convolution operations have been replaced by Hankelization in pre-processing and post-processing. Compared with ConvNet, the forward and backward embedding operations can be implemented by convolution and transposed convolution with one-hot-filters (see Fig. 12 in Appendix for details). Note that the encoder-decoder part can be implemented by multiple convolution layers with kernel size (1,1) and non-linear activations. In our model, we do not use convolution explicitly but just do linear transform and non-linear activation for "filter-domain" (*i.e.,* horizontal axis of tensors in Fig. 1).

The contributions in this study can be summarized as follow: (1) A new and simple approach of image/tensor modeling is proposed which translates the ConvNet, (2) effectiveness of the proposed method and similarity to the DIP are demonstrated in experiments, and (3) most importantly, there is a prospect for interpreting/characterizing the DIP as "low-dimensional patch-manifold prior".

## 2 RELATED WORKS

Note that the idea of low-dimensional patch manifold itself has been proposed by Peyre (2009) and Osher et al. (2017). Peyre had firstly formulated the patch manifold model of natural images and solve it by dictionary learning and manifold pursuit. Osher *et al.* formulated the regularization function to minimize dimension of patch manifold, and solved Laplace-Beltrami equation by point integral method. In comparison with these studies, we decrease the dimension of patch-manifold by utilizing AE shown in Fig. 1.

A related technique, low-rank tensor modeling in embedded space, has been studied recently by Yokota et al. (2018). However, the modeling approaches here are different: multi-linear vs non-linear manifold. Thus, our study would be interpreted as manifold version of (Yokota et al., 2018) in a perspective of tensor completion methods. Note that Yokota et al. (2018) applied their model for only tensor completion task. By contrast, we investigate here tensor completion, super-resolution, and deconvolution tasks.

Another related work is devoted to group sparse representation (GSR) (Zhang et al., 2014a). The GSR is roughly characterized as a combination of similar patch-grouping and sparse modeling which is similar to the combination of embedding and manifold-modeling. However, the computational cost of similar patch-grouping is obviously higher than embedding, and this task is naturally included in manifold learning.

The main difference between above studies and our is the motivation: Essential and simple image modeling which can translate the ConvNet/DIP. The proposed MMES has many connections with ConvNet/DIP such as embedding, non-linear mapping, and the training with noise.

From a perspective of DIP, there are several related works. First, the deep geometric prior (Williams et al., 2019) utilises a good properties of a multi-layer perceptron for shape reconstruction problem which efficiently learn a smooth function from 2D space to 3D space. It helps us to understand DIP from a perspective of manifold learning. For example, it can be used for gray scale image reconstruction if an image is regarded as point could in 3D space $(i, j, X_{ij})$. However, this may not provide the good image reconstruction like DIP, because it just smoothly interpolates a point cloud by surface like a Volonoi interpolation. Especially it can not provide a property of self-similarity in natural image.

Second, deep decoder (Heckel & Hand, 2018) reconstructs natural images from noises by non-convolutional networks which consists of linear channel/color transform, ReLU, channel/color normalization, and upsampling layers. In contrast that DIP uses over-parameterized network, deep decoder uses under-parameterized network and shows its ability of image reconstruction. Although deep decoder is a non-convolutional network, Authors emphasize the closed relationship between convolutional layers in DIP and upsampling layers in deep decoder. In this literature, Authors described "If there is no upsampling layer, then there is no notion of locality in the resultant image" in deep decoder. It implies the "locality" is the essence of image model, and the convolution/upsampling layer provides it. Furthermore, the deep decoder has a close relationship with our MMES. Note that the MMES is originally/essentially has only decoder and inverse MDT (see Eq.(3)), and the encoder is just used for satisfying Hankel structure. The decoder and inverse MDT in our MMES are respectively corresponding linear operation and upsampling layer in deep decoder. Moreover, concept of under-parameterization is also similar to our MMES.

From this, we can say the essence of image model is the "locality", and its locality can be provided by "convolution", "upsampling", or "delay-embedding". This is why the image restoration from single image with deep convolutional networks has highly attentions which are called by zero-shot learning, internal learning, or self-supervised learning (Shocher et al., 2018; Lehtinen et al., 2018; Krull et al., 2019; Batson & Royer, 2019; Xu et al., 2019; Cha et al., 2019; Laine et al., 2019).

Recently, two generative models: SinGAN (Shaham et al., 2019) and InGAN (Shocher et al., 2019) learned from only a single image, have been proposed. Key concept of both papers is to impose the constraint for local patches of image to be natural. From a perspective of the constraint for local patches of image, our MMES has closed relationship with these works. However, we explicitly impose a low-dimensional manifold constraint for local patches rather than adversarial training with patch discriminators.

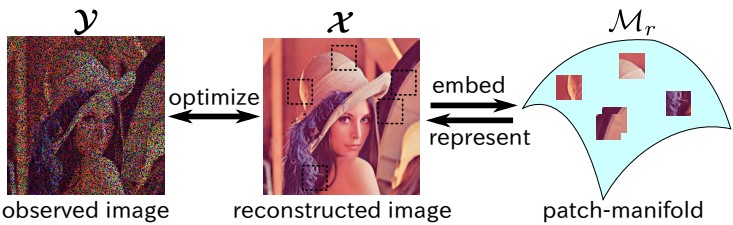

Figure 2: Conceptual illustration of MMES for a image inpainting task.

# 3 MANIFOLD MODELING IN EMBEDDED SPACE

Here, on the contrary to Section 1, we start to explain the proposed method from the concept of MMES, and we systematically derive the MMES structure from it. Conceptually, the proposed tensor reconstruction method can be formulated by

$$\underset{\mathcal{X}}{\text{minimize}} \; ||\mathcal{Y} - \mathcal{F}(\mathcal{X})||_F^2,$$
$$\text{s.t.} \; \mathcal{H}(\mathcal{X}) = [\boldsymbol{h}_1, \boldsymbol{h}_2, ..., \boldsymbol{h}_T] =: \boldsymbol{H}, \tag{1}$$
$$\boldsymbol{h}_t \in \mathcal{M}_r \; \text{for} \; t = 1, 2, ..., T,$$

where $\mathcal{Y} \in \mathbb{R}^{J_1 \times J_2 \times \cdots \times J_N}$ is an observed corrupted tensor, $\mathcal{X} \in \mathbb{R}^{I_1 \times I_2 \times \cdots \times I_N}$ is an estimated tensor, $\mathcal{F} : \mathbb{R}^{I_1 \times I_2 \times \cdots \times I_N} \to \mathbb{R}^{J_1 \times J_2 \times \cdots \times J_N}$ is a linear operator which represents the observation system, $\mathcal{H} : \mathbb{R}^{I_1 \times I_2 \times \cdots \times I_N} \to \mathbb{R}^{D \times T}$ is padding and Hankelization operator with sliding window of size $(\tau_1, \tau_2, ..., \tau_N)$, and we impose each column of matrix $\boldsymbol{H}$ can be sampled from an $r$-dimensional manifold $\mathcal{M}_r$ in $D$-dimensional Euclid space (see Appendix B for details). We have $r \leq D$. For simplicity, we putted $D := \prod_n \tau_n$ and $T := \prod_n (I_n + \tau_n - 1)$. For tensor completion task, $\mathcal{F} := P_\Omega$ is a projection operator onto support set $\Omega$ so that the missing elements are set to be zero. For super-resolution task, $\mathcal{F}$ is a down-sampling operator of images/tensors. For deconvolution task, $\mathcal{F}$ is a convolution operator with some blur kernels. Fig. 2 shows the concept of proposed manifold modeling in case of image inpainting (*i.e.*, $N = 2$). We minimize the distance between observation $\mathcal{Y}$ and reconstruction $\mathcal{X}$ with its support $\Omega$, and all patches in $\mathcal{X}$ should be included in some restricted manifold $\mathcal{M}_r$. In other words, $\mathcal{X}$ is represented by the patch-manifold, and the property of the patch-manifold can be image priors. For example, low dimensionality of patch-manifold restricts the non-local similarity of images/tensors, and it would be related with "impedance" in DIP. We model $\mathcal{X}$ indirectly by designing the properties of patch-manifold $\mathcal{M}_r$.

## 3.1 DEFINITION OF LOW-DIMENSIONAL MANIFOLD

We consider an AE to define the $r$-dimensional manifold $\mathcal{M}_r$ in $(\prod_n \tau_n)$-dimensional Euclidean space as follows:

$$\mathcal{M}_r := \{\hat{\psi}_r(\boldsymbol{l}) \mid \boldsymbol{l} \in \mathbb{R}^r\}, \; (\hat{\psi}_r, \hat{\phi}_r) := \underset{(\psi_r, \phi_r)}{\text{argmin}} \sum_{t=1}^{T} ||\boldsymbol{h}_t - \psi_r \phi_r(\boldsymbol{h}_t)||_2^2, \tag{2}$$

where $\phi_r : \mathbb{R}^D \to \mathbb{R}^r$ is an encoder, $\psi_r : \mathbb{R}^r \to \mathbb{R}^D$ is a decoder, and $\hat{\psi}_r \hat{\phi}_r : \mathbb{R}^D \to \mathbb{R}^D$ is an auto-encoder constructed from $\{\boldsymbol{h}_t\}_{t=1}^T$. Note that, in general, the use of AE models is a widely accepted approach for manifold learning (Hinton & Salakhutdinov, 2006). The properties of the manifold $\mathcal{M}_r$ are determined by the properties of $\phi_r$ and $\psi_r$. By employing multi-layer perceptrons (neural networks) for $\phi_r$ and $\psi_r$, encoder-decoder may provide a smooth manifold.

## 3.2 PROBLEM FORMULATION

In this section, we combine the conceptual formulation (1) and the AE guided manifold constraint to derive a equivalent and more practical optimization problem. First, we redefine a tensor $\mathcal{X}$ as an output of generator:

$$\mathcal{X} := \mathcal{H}^\dagger [\boldsymbol{h}_1, \boldsymbol{h}_2, ..., \boldsymbol{h}_T], \quad \text{where} \; \boldsymbol{h}_t \in \mathcal{M}_r$$
$$= \mathcal{H}^\dagger [\hat{\psi}_r(\boldsymbol{l}_1), \hat{\psi}_r(\boldsymbol{l}_2), ..., \hat{\psi}_r(\boldsymbol{l}_T)], \tag{3}$$

---

**Algorithm 1** Optimization algorithm for tensor reconstruction
___
    **input**: $\boldsymbol{\mathcal{Y}} \in \mathbb{R}^{J_1 \times \cdots \times J_N}$ (corrupted tensor), $\mathcal{F}, \boldsymbol{\tau}, r, \sigma$;
    **initialize**: $\boldsymbol{\mathcal{Z}} \in \mathbb{R}^{I_1 \times \cdots \times I_N}$, auto-encoder $\mathcal{A}_r, \lambda = 5.0$;
    **repeat**
        $\boldsymbol{H} \leftarrow \mathcal{H}(\boldsymbol{\mathcal{Z}}) \in \mathbb{R}^{D \times T}$ with $\boldsymbol{\tau}$;
        generate noise $\boldsymbol{E} \in \mathbb{R}^{D \times T}$ with $\sigma$;
        $\mathcal{L}_{\text{AE}} \leftarrow ||\boldsymbol{H} - \mathcal{A}_r(\boldsymbol{H} + \boldsymbol{E})||_F^2$;
        $\mathcal{L}_{\text{rec}} \leftarrow \frac{1}{D}||\boldsymbol{\mathcal{Y}} - \mathcal{F}(\mathcal{H}^\dagger \mathcal{A}_r(\boldsymbol{H} + \boldsymbol{E}))||_F^2$;
        update $(\boldsymbol{\mathcal{Z}}, \mathcal{A}_r)$ by $\mathtt{Adam}$ for $\mathcal{L}_{\text{rec}} + \lambda \mathcal{L}_{\text{AE}}$;
        **if** $\mathcal{L}_{\text{rec}} < \mathcal{L}_{\text{AE}}$ **then** $\lambda \leftarrow 1.1\lambda$; **else** $\lambda \leftarrow 0.99\lambda$;
    **until** converge
    **output**: $\widehat{\boldsymbol{\mathcal{X}}} = \mathcal{H}^\dagger \mathcal{A}_r \mathcal{H}(\boldsymbol{\mathcal{Z}}) \in \mathbb{R}^{I_1 \times \cdots \times I_N}$ (reconstructed tensor);
___

where $\boldsymbol{l}_t \in \mathbb{R}^r$, and $\mathcal{H}^\dagger$ is a pseudo inverse of $\mathcal{H}$. At this moment, $\boldsymbol{\mathcal{X}}$ is a function of $\{\boldsymbol{l}_t\}_{t=1}^T$, however Hankel structure of matrix $\boldsymbol{H}$ can not be always guaranteed under the unconstrained condition of $\boldsymbol{l}_t$. For guaranteeing the Hankel structure of matrix $\boldsymbol{H}$, we further transform it as follow:

$$
\begin{aligned}
\boldsymbol{\mathcal{X}} :&= \mathcal{H}^\dagger[\hat{\psi}_r \hat{\phi}_r(\boldsymbol{g}_1), \hat{\psi}_r \hat{\phi}_r(\boldsymbol{g}_2), ..., \hat{\psi}_r \hat{\phi}_r(\boldsymbol{g}_T)], \\
&= \mathcal{H}^\dagger \mathcal{A}_r[\boldsymbol{g}_1, \boldsymbol{g}_2, ..., \boldsymbol{g}_T] \\
&= \mathcal{H}^\dagger \mathcal{A}_r \mathcal{H}(\boldsymbol{\mathcal{Z}}),
\end{aligned}
\tag{4}
$$

where we put $\mathcal{A}_r : \mathbb{R}^{D \times T} \rightarrow \mathbb{R}^{D \times T}$ as an operator which auto-encodes each column of a input matrix with $(\hat{\psi}_r, \hat{\phi}_r)$, and $[\boldsymbol{g}_1, \boldsymbol{g}_2, ..., \boldsymbol{g}_T]$ as a matrix, which has Hankel structure and is transformed by Hankelization of some input tensor $\boldsymbol{\mathcal{Z}} \in \mathbb{R}^{I_1 \times I_2 \times \cdots \times I_N}$. Note that $\boldsymbol{\mathcal{Z}}$ is the most compact representation for Hankel matrix $[\boldsymbol{g}_1, \boldsymbol{g}_2, ..., \boldsymbol{g}_T]$. Eq. (4) describes the MMES network shown in Fig. 1: $\mathcal{H}, \hat{\phi}_r, \hat{\psi}_r$ and $\mathcal{H}^\dagger$ are respectively corresponding to forward embedding, encoding, decoding, and backward embedding, where encoder and decoder can be defined *e.g.* by multi-layer perceptrons (*i.e.,* repetition of linear transformation and non-linear activation).

From this formulation, Problem (1) is transformed as minimize$_{\boldsymbol{\mathcal{Z}}} ||\boldsymbol{\mathcal{Y}} - \mathcal{F}(\mathcal{H}^\dagger \mathcal{A}_r \mathcal{H}(\boldsymbol{\mathcal{Z}}))||_F^2$, where $\mathcal{A}_r$ is an AE which defines the manifold $\mathcal{M}_r$. In this study, the AE/manifold is learned from an observed tensor $\boldsymbol{\mathcal{Y}}$ itself, thus the optimization problem is finally formulated as

$$
\underset{\boldsymbol{\mathcal{Z}}, \mathcal{A}_r}{\text{minimize}} \underbrace{||\boldsymbol{\mathcal{Y}} - \mathcal{F}(\mathcal{H}^\dagger \mathcal{A}_r \mathcal{H}(\boldsymbol{\mathcal{Z}}))||_F^2}_{=:\mathcal{L}_{\text{rec}}} + \lambda \underbrace{||\mathcal{H}(\boldsymbol{\mathcal{Z}}) - \mathcal{A}_r \mathcal{H}(\boldsymbol{\mathcal{Z}})||_F^2}_{=:\mathcal{L}_{\text{AE}}},
\tag{5}
$$

where we refer respectively the first and second terms by a reconstruction loss and an auto-encoding loss, and $\lambda > 0$ is a trade-off parameter for balancing both losses.

### 3.3 OPTIMIZATION ALGORITHM

Optimization problem (5) consists of two terms: a reconstruction loss, and an auto-encoding loss. Hyperparameter $\lambda$ is set to balance both losses. Basically, $\lambda$ should be large because auto-encoding loss should be zero. However, very large $\lambda$ prohibits minimizing the reconstruction loss, and may lead to local optima. Therefore, we adjust gradually the value of $\lambda$ in the optimization process.

Algorithm 1 shows an optimization algorithm for tensor reconstruction and/or enhancement. For AE learning, we employs a strategy of denoising-auto-encoder (see Appendix in detail). Adaptation of $\lambda$ is just an example, and it can be modified appropriately with data. Here, the trade-off parameter $\lambda$ is adjusted for keeping $\mathcal{L}_{\text{rec}} > \mathcal{L}_{\text{AE}}$, but for no large gap between both losses. By exploiting the convolutional structure of $\mathcal{H}$ and $\mathcal{H}^\dagger$ (see Appendix B.1), the calculation flow of $\mathcal{L}_{\text{rec}}$ and $\mathcal{L}_{\text{AE}}$ can be easily implemented by using neural network libraries such as $\mathtt{TensorFlow}$. We employed $\mathtt{Adam}$ (Kingma & Ba, 2014) optimizer for updating $(\boldsymbol{\mathcal{Z}}, \mathcal{A}_r)$.

## 4 EXPERIMENTS

Here, we show the selective experimental results to demonstrate the close similarity and some slight differences between DIP and MMES. First, toy examples with a time-series signal and a gray-scale

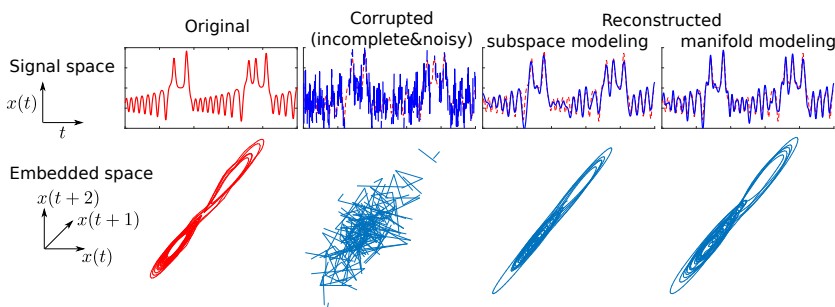

Figure 3: Time series signal recovery of subspace and manifold models in embedded space.

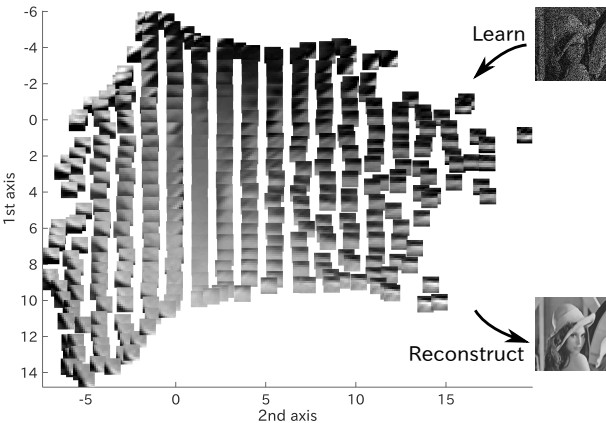

Figure 4: Distribution of two-dimensional (8,8)-patches on manifold learned from a 50% missing gray-scale image of 'Lena'.

image were recovered by the proposed method to show its basic behaviors. Thereafter, we show the main results by comparison with DIP and other selective methods on color-image inpainting, super-resolution, and deconvolution tasks. Optional results of optimization behavior, hyper-parameter sensitivity, and volumetric/3D image completion are shown in Appendix.

## 4.1 TOY EXAMPLES

In this section, we apply the proposed method into a toy example of signal recovery. Fig. 3 shows a result of this experiment. A one-dimensional time-series signal is generated from Lorentz system, and corrupted by additive Gaussian noise, random missing, and three block occlusions. The corrupted signal was recovered by the subspace modeling (Yokota et al., 2018), and the proposed manifold modeling in embedded space. Window size of delay-embedding was $\tau = 64$, the lowest dimension of auto-encoder was $r = 3$, and additive noise standard deviation was set to $\sigma = 0.05$. Manifold modeling catched the structure of Lorentz attractor much better than subspace modeling.

Fig. 4 visualizes a two-dimensional $(8, 8)$-patch manifold learned by the proposed method from a 50% missing gray-scale image of 'Lena'. For this figure, we set $\tau = [8, 8]$, $r = 2$, $\sigma = 0.05$. Similar patches are located near each other, and the smooth change of patterns can be observed. It implies the relationship between non-local similarity based methods (Buades et al., 2005; Dabov et al., 2007; Gu et al., 2014; Zhang et al., 2014a), and the manifold modeling (*i.e.,* DAE) plays a key role of "patch-grouping" in the proposed method. The difference from the non-local similarity based approach is that the manifold modeling is "global" rather than "non-local" which finds similar patches of the target patch from its neighborhood area.

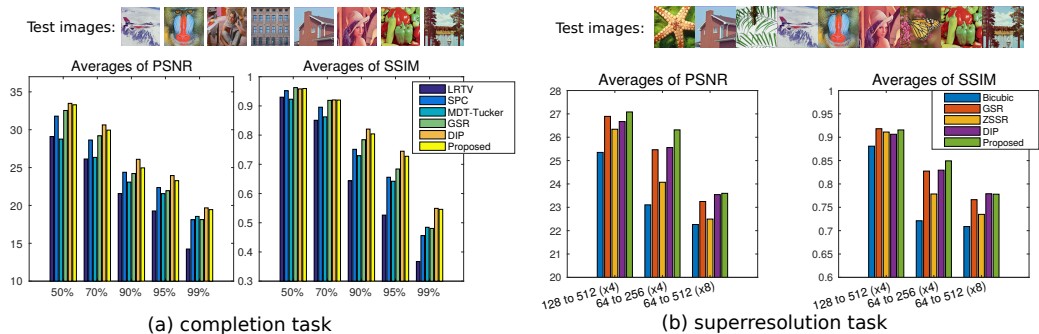

Figure 5: Comparison of performance by averages of PSNR and SSIM for color image completion and super-resolution tasks with various settings.

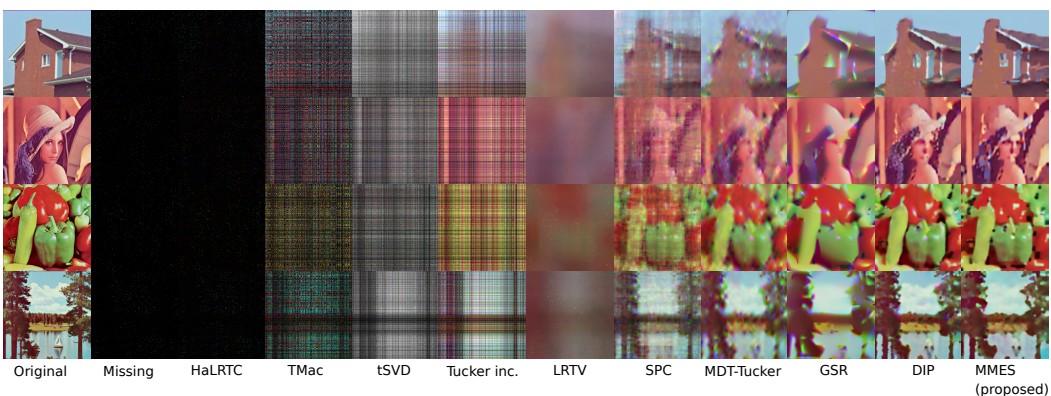

Figure 6: Illustration of image inpainting results from nature images with 99% missing pixels by HaLTRC, TMac, tSVD, Tucker inc., LRTV, SPC, GSR, MDT-Tucker, DIP and the proposed MMES. Note that DIP and MMES approaches provide the best performance in comparison to state-of-the-arts methods.

## 4.2 COLOR IMAGE COMPLETION, ESPECIALLY FOR EXTREMELY HIGH NUMBER OF MISSING PIXELS

In this section, we compare performance of the proposed method with several selected unsupervised image inpainting methods: low-rank tensor completion (HaLRTC) (Liu et al., 2013), parallel low-rank matrix factorization (TMac) (Xu et al., 2015), tubal nuclear norm regularization (tSVD) (Zhang et al., 2014b), Tucker decomposition with rank increment (Tucker inc.) (Yokota et al., 2018), low-rank and total-variation (LRTV) regularization (Yokota & Hontani, 2017; 2019), smooth PARAFAC tensor completion (SPC) (Yokota et al., 2016), GSR (Zhang et al., 2014a), multi-way delay embedding based Tucker modeling (MDT-Tucker) (Yokota et al., 2018), and DIP (Ulyanov et al., 2018). Implementation and detailed hyper-parameter settings are explained in Appendix. Basically, we carefully tuned the hyper-parameters for all methods to perform the best scores of peak-signal-to-noise ratio (PSNR) and structural similarity (SSIM).

Fig. 5(a) shows the eight test images and averages of PSNR and SSIM for various missing ratio {50%, 70%, 90%, 95%, 99%} and for selective competitive methods. The proposed method is quite competitive with DIP. Fig. 6 shows the illustration of results. The 99% of randomly selected voxels are removed from 3D (256,256,3)-tensors, and the tensors were recovered by various methods. Basically low-rank priors (HaLRTC, TMac, tSVD, Tucker) could not recover such highly incomplete image. In piecewise smoothness prior (LRTV), over-smoothed images were reconstructed since the essential image properties could not be captured. There was a somewhat jump from them by SPC (i.e., smooth prior of basis functions in low-rank tensor decomposition). MDT-Tucker further improves it by exploiting the shift-invariant multi-linear basis. GSR nicely recovered the global

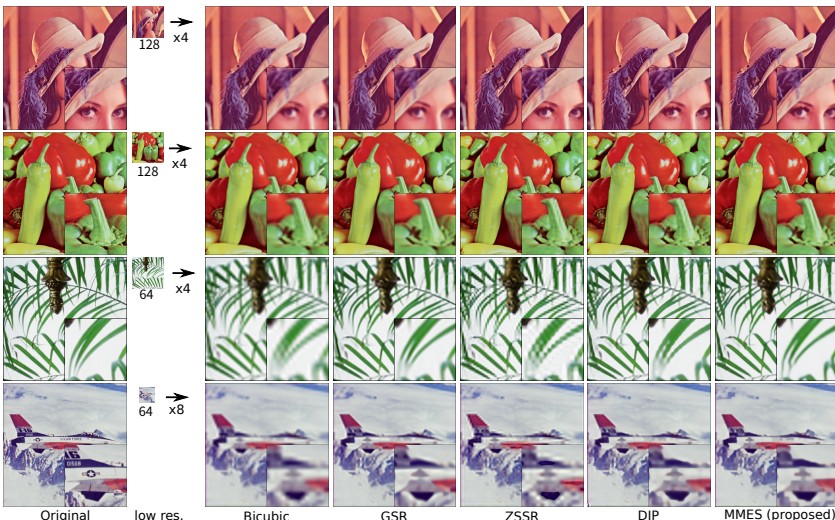

Figure 7: Comparison of our approach with other methods for super-resolution task. The first line 'Leaves' were up-scaled from (64,64,3) to (256,256,3), and the second line 'Airplane' was up-scaled from (64,64,3) to (512,512,3).

pattern of images but details were insufficient. Finally, the reconstructed images by DIP and MMES recovered both global and local patterns of images.

### 4.3 COLOR IMAGE SUPER-RESOLUTION

In this section, we compare the proposed method with selected unsupervised image super-resolution methods: Bicubic interpolation, GSR (Zhang et al., 2014a), ZSSR (Shocher et al., 2018) and DIP (Ulyanov et al., 2018). Implementation and detailed hyper-parameter settings are explained in Appendix. Basically, we carefully tuned the hyper-parameters for all methods to perform the best scores of PSNR and SSIM.

Fig. 5(b) shows values of PSNR and SSIM of the computer simulation results. We used three (256,256,3) color images, and six (512,512,3) color images. Super resolution methods scaling up them from four or eight times down-scaled images of them with Lanczos2 kernels. According to this quantitative evaluation, bicubic interpolation was clearly worse than others. ZSSR worked well for up-scaling from (128,128,3), however the performances were substantially decreased for up-scaling from (64,64,3). Basically, GSR, DIP, and MMES were very competitive. In detail, DIP was slightly better than GSR, and the proposed MMES was slightly better than DIP. More detailed PSNR/SSIM values are given by Table 3 in Appendix. Fig. 7 shows selected high resolution images reconstructed by four super-resolution methods. In general, bicubic method reconstructed blurred images and these were visually worse than others. GSR results had smooth outlines in all images, but these were slightly blurred. ZSSR was weak for very low-resolution images. DIP reconstructed visually sharp images but these images had jagged artifacts along the diagonal lines. The proposed MMES reconstructed sharp and smooth outlines.

### 4.4 COLOR IMAGE DECONVOLUTION

In this section, we compare the proposed method with DIP for image deconvolution/deblurring task. Three (256,256,3) color images are prepared and blurred by using three different Gaussian filters. For DIP we choose the best early stopping timing from {1000, 2000, ..., 10000} iterations. For MMES, we employed the fixed AE structure as $[32\tau^2, r, 32\tau^2]$, and parameters as $\tau = 4$, $r = 16$, and $\sigma = 0.01$ for all nine cases. Fig. 8 shows the reconstructed deblurring images by DIP and MMES. Tab. 1 shows the PSNR and SSIM values of these results. We can see that the similarity of the methods qualitatively and quantitatively.

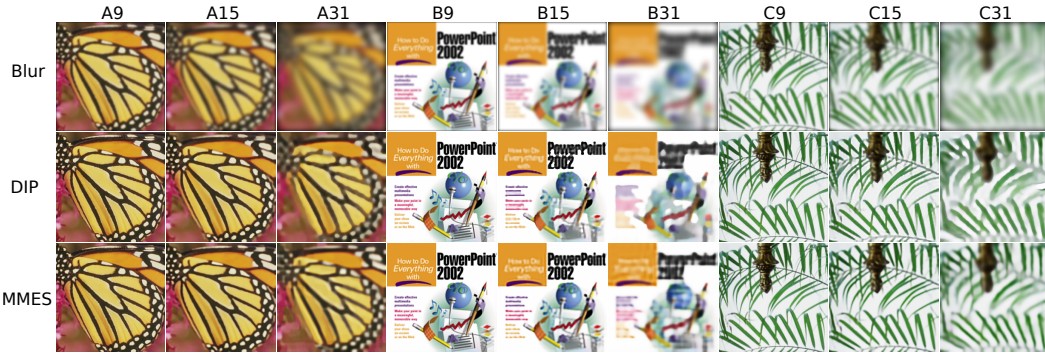

Figure 8: Comparison of our approach with DIP for deconvolution/deblurring task. Three color images were blurred by three Gaussian windows of different sizes. These were recovered by the DIP and the proposed MMES.

Table 1: PSNR/SSIM values for deconvolution results

| PSNR | A9 | A15 | A30 | B9 | B15 | B31 | C9 | C15 | C31 |
|------|------|------|------|------|------|------|------|------|------|
| DIP | 28.85 | 24.53 | 18.21 | 26.58 | 19.80 | 16.21 | 27.18 | 22.07 | 15.74 |
| MMES | **29.24** | **24.72** | **18.92** | **27.46** | **20.89** | **16.45** | **29.15** | **22.38** | **16.55** |
| SSIM | A9 | A15 | A30 | B9 | B15 | B31 | C9 | C15 | C31 |
| DIP | .9346 | .8586 | .5962 | .9438 | .7871 | .5628 | .9438 | .8276 | .4568 |
| MMES | **.9436** | **.8599** | **.6234** | **.9512** | **.8088** | **.5805** | **.9636** | **.8382** | **.5284** |

# 5 INTERPRETATION OF MMES TOWARD EXPLAINING DIP

It is well known that there is no mathematical definition of interpretability in machine learning and there is no one unique definition of interpretation. We understand the interpretability as a degree to which a human can consistently predict the model's results or performance. The higher the interpretability of a deep learning model, the easier it is for someone to comprehend why certain performance or predictions or expected output can be achieved. We think that a model is better interpretable than another model if its performance or behaviors are easier for a human to comprehend than performance of the other models.

## 5.1 FROM A PERSPECTIVE OF DIMENSIONALITY REDUCTION/MANIFOLD LEARNING

The manifold learning and associated auto-encoder (AE) can be viewed as the generalized non-linear version of principal component analysis (PCA). In fact, manifold learning solves the key problem of dimensionality reduction very efficiently. In other words, manifold learning (modeling) is an approach to non-linear dimensionality reduction. Manifold modeling for this task are based on the idea that the dimensionality of many data sets is only artificially high. Although the patches of images (data points) consist of hundreds/thousands pixels, they may be represented as a function of only a few or quite limited number underlying parameters. That is, the patches are actually samples from a low-dimensional manifold that is embedded in a high-dimensional space. Manifold learning algorithms attempt to uncover these parameters in order to find a low dimensional representation of the images.

In our MMES approach to solve the problem we applied original embedding via multi-way delay embedding transform (MDT or Hankelization). Our algorithm is based on the optimization of cost function and it works towards extracting the low-dimensional manifold that is used to describe the high-dimensional data. The manifold is described mathematically by Eq. (2) and cost function is formulated by Eq. (5).

## 5.2 REGARDING OUR ATTEMPT TO INTERPRET "NOISE IMPEDANCE IN DIP" VIA MMES

As mentioned at introduction, Ulyanov et al. (2018) reported an important phenomenon of noise impedance of ConvNet structures. Here, we provide a prospect for explaining the noise impedance in DIP through the MMES.

Let us consider the sparse-land model, i.e. noise-free images are distributed along low-dimensional manifolds in the high-dimensional Euclidean space and images perturbed by noises thicken the manifolds (make the dimension of the manifolds higher). Under this model, the distribution of images can be assumed to be higher along the low-dimensional noise-free image manifolds. When we assume that the image patches are sampled from low-dimensional manifold like sparse-land model, it is difficult to put noisy patches on the low-dimensional manifold. Let us consider to fit the network for noisy images. In such case the fastest way for decreasing squared error (loss function) is to learn "similar patches" which often appear in a large set of image-patches. Note that finding similar image-patches for denoising is well-known problem solved, e.g., by BM3D algorithm, which find similar image patches by template matching. In contrast, our auto-encoder automatically maps similar-patches into close points on the low-dimensional manifold. When similar-patches have some noise, the low-dimensional representation tries to keep the common components of similar patches, while reducing the noise components. This has been proved by Alain & Bengio (2014) so that a (denoising) auto-encoder maps input image patches toward higher density portions in the image space. In other words, a (denoising) auto-encoder has kind of a force to reconstruct the low-dimensional patch manifold, and this is our rough explanation of noise impedance phenomenon. Although the proposed MMES and DIP are not completely equivalent, we see many analogies and similarities and we believe that our MMES model and associated learning algorithm give some new insight for DIP.

## 6 DISCUSSIONS AND CONCLUSIONS

A beautiful manifold representation of complicated signals in embedded space has been originally discovered in a study of dynamical system analysis (*i.e.,* chaos analysis) for time-series signals (Packard et al., 1980). After this, many signal processing and computer vision applications have been studied but most methods have considered only linear approximation because of the difficulty of non-linear modeling (Van Overschee & De Moor, 1991; Szummer & Picard, 1996; Li et al., 1997; Ding et al., 2007; Markovsky, 2008). However nowadays, the study of non-linear/manifold modeling has been well progressed with deep learning, and it was successfully applied in this study. Interestingly, we could apply this non-linear system analysis not only for time-series signals but also natural color images and tensors (this is an extension from delay-embedding to multi-way shift-embedding). The best of our knowledge, this is the first study to apply Hankelization with AE into general tensor data reconstruction.

MMES is a novel and simple image reconstruction model based on the low-dimensional patch-manifold prior which has many connections to ConvNet. We believe it helps us to understand how work ConvNet/DIP through MMES, and support to use DIP for various applications like tensor/image reconstruction or enhancement (Gong et al., 2018; Yokota et al., 2019; Van Veen et al., 2018; Gandelsman et al., 2019).

Finally, we established bridges between quite different research areas such as the dynamical system analysis, the deep learning, and the tensor modeling. The proposed method is just a prototype and can be further improved by incorporating other methods such as regularizations, multi-scale extensions, and adversarial training.

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

## A    HANKELIZATION OF ONE- AND TWO-DIMENSIONAL ARRAYS

For example, Hankelization of one-dimensional array $\boldsymbol{f} = [f_1, f_2, ..., f_7]$ with window size $\tau = 3$ is given by

$$
\begin{pmatrix}
f_1 & f_2 & f_3 & f_4 & f_5 \\
f_2 & f_3 & f_4 & f_5 & f_6 \\
f_3 & f_4 & f_5 & f_6 & f_7
\end{pmatrix}. \tag{6}
$$

We can see the anti-diagonal elements of above matrix are equivalent. Such matrix is called as "Hankel matrix".

For a two-dimensional array

$$
\begin{pmatrix}
f_{11} & f_{12} & f_{13} \\
f_{21} & f_{22} & f_{23} \\
f_{31} & f_{32} & f_{33}
\end{pmatrix}, \tag{7}
$$

we consider unfold of it and inverse folding by

$$
\mathrm{unfold}\begin{pmatrix}
f_{11} & f_{12} & f_{13} \\
f_{21} & f_{22} & f_{23} \\
f_{31} & f_{32} & f_{33}
\end{pmatrix} = \begin{pmatrix} f_{11} \\ f_{21} \\ f_{31} \\ f_{12} \\ f_{22} \\ f_{32} \\ f_{13} \\ f_{23} \\ f_{33} \end{pmatrix}, \text{ and } \begin{pmatrix}
f_{11} & f_{12} & f_{13} \\
f_{21} & f_{22} & f_{23} \\
f_{31} & f_{32} & f_{33}
\end{pmatrix} = \mathrm{fold}\begin{pmatrix} f_{11} \\ f_{21} \\ f_{31} \\ f_{12} \\ f_{22} \\ f_{32} \\ f_{13} \\ f_{23} \\ f_{33} \end{pmatrix}. \tag{8}
$$

The point here is that we scan matrix elements column-wise manner. Hankelization of this two-dimensional array (matrix) with $\boldsymbol{\tau} = [2, 2]$ is given by scanning a matrix with local (2,2)-window column-wise manner, and unfold and stack each local patch left-to-right. Thus, it is given as

$$
\left( \begin{pmatrix} f_{11} \\ f_{21} \\ f_{12} \\ f_{22} \end{pmatrix} \begin{pmatrix} f_{21} \\ f_{31} \\ f_{22} \\ f_{32} \end{pmatrix} \begin{pmatrix} f_{12} \\ f_{22} \\ f_{13} \\ f_{23} \end{pmatrix} \begin{pmatrix} f_{22} \\ f_{32} \\ f_{23} \\ f_{33} \end{pmatrix} \right) = \left( \begin{pmatrix} f_{11} & f_{21} \\ f_{21} & f_{31} \\ f_{12} & f_{22} \\ f_{22} & f_{32} \end{pmatrix} \begin{pmatrix} f_{12} & f_{22} \\ f_{22} & f_{32} \\ f_{13} & f_{23} \\ f_{23} & f_{33} \end{pmatrix} \right). \tag{9}
$$

We can see that it is *not* a Hankel matrix. However, it is a "block Hankel matrix" in perspective of block matrix, a matrix that its elements are also matrices. We can see the block matrix itself is a Hankel matrix and all elements are Hankel matrices, too. Thus, Hankel matrix is a special case of block Hankel matrix in case of that all elements are scalar. In this paper, we say simply "Hankel structure" for block Hankel structure.

Figure 9 shows an illustrative explanation of valid convolution which is decomposed into delay-embedding/Hankelization and linear transformation. 1D valid convolution of $\boldsymbol{f}$ with kernel $\boldsymbol{h} = [h_1, h_2, h_3]$ can be provided by matrix-vector product of the Hankel matrix and $\boldsymbol{h}$. In similar way, 2D valid convolution can be provided by matrix-vector product of the block Hankel matrix and unfolded kernel.

## B    MULTIWAY-DELAY EMBEDDING FOR TENSORS

Multiway-delay embedding transform (MDT) is a multi-way generalization of Hankelization proposed by Yokota et al. (2018).

In (Yokota et al., 2018), MDT is defined by using the multi-linear tensor product with multiple duplication matrices and tensor reshaping. Basically, we use the same operation, but a padding operation is added. Thus, the multiway-delay embedding used in this study is defined by

$$
\mathcal{H}(\boldsymbol{\mathcal{X}}) := \mathrm{unfold}_{(D,T)}(\mathrm{pad}_{\boldsymbol{\tau}}(\boldsymbol{\mathcal{X}}) \times_1 \boldsymbol{S}_1 \cdots \times_N \boldsymbol{S}_N), \tag{10}
$$

$$
\text{(1D case)} \quad
\begin{bmatrix} f_1 \\ f_2 \\ f_3 \\ f_4 \\ f_5 \\ f_6 \\ f_7 \end{bmatrix}
*
\begin{bmatrix} h_1 \\ h_2 \\ h_3 \end{bmatrix}
=
\begin{bmatrix}
f_1 h_1 + f_2 h_2 + f_3 h_3 \\
f_2 h_1 + f_3 h_2 + f_4 h_3 \\
f_3 h_1 + f_4 h_2 + f_5 h_3 \\
f_4 h_1 + f_5 h_2 + f_6 h_3 \\
f_5 h_1 + f_6 h_2 + f_7 h_3
\end{bmatrix}
=
\begin{bmatrix}
f_1 & f_2 & f_3 \\
f_2 & f_3 & f_4 \\
f_3 & f_4 & f_5 \\
f_4 & f_5 & f_6 \\
f_5 & f_6 & f_7
\end{bmatrix}
\times
\begin{bmatrix} h_1 \\ h_2 \\ h_3 \end{bmatrix}
$$

Delay-embedding
or Hankelization
+ Linear transform

$$
\text{(2D case)} \quad
\begin{bmatrix}
f_{11}\ f_{12}\ f_{13} \\
f_{21}\ f_{22}\ f_{23} \\
f_{31}\ f_{32}\ f_{33}
\end{bmatrix}
*
\begin{bmatrix}
h_{11}\ h_{12} \\
h_{21}\ h_{22}
\end{bmatrix}
=
\begin{bmatrix}
\begin{bmatrix} f_{11}\ f_{12} \\ f_{21}\ f_{22} \end{bmatrix} \cdot \begin{bmatrix} h_{11}\ h_{12} \\ h_{21}\ h_{22} \end{bmatrix} &
\begin{bmatrix} f_{12}\ f_{13} \\ f_{22}\ f_{23} \end{bmatrix} \cdot \begin{bmatrix} h_{11}\ h_{12} \\ h_{21}\ h_{22} \end{bmatrix} \\[2em]
\begin{bmatrix} f_{21}\ f_{22} \\ f_{31}\ f_{32} \end{bmatrix} \cdot \begin{bmatrix} h_{11}\ h_{12} \\ h_{21}\ h_{22} \end{bmatrix} &
\begin{bmatrix} f_{22}\ f_{23} \\ f_{32}\ f_{33} \end{bmatrix} \cdot \begin{bmatrix} h_{11}\ h_{12} \\ h_{21}\ h_{22} \end{bmatrix}
\end{bmatrix}
$$

$$
= \text{fold}
\begin{bmatrix}
f_{11}\ f_{21}\ f_{12}\ f_{22} \\
f_{21}\ f_{31}\ f_{22}\ f_{32} \\
f_{12}\ f_{22}\ f_{13}\ f_{23} \\
f_{22}\ f_{32}\ f_{23}\ f_{33}
\end{bmatrix}
\times
\begin{bmatrix} h_{11} \\ h_{21} \\ h_{12} \\ h_{22} \end{bmatrix}
$$

Figure 9: Decomposition of 1D and 2D convolutions: Valid convolution can be divided into delay-embedding/Hankelization and linear transformation.

where $\text{pad}_{\boldsymbol{\tau}} : \mathbb{R}^{I_1 \times \cdots \times I_N} \to \mathbb{R}^{(I_1 + 2(\tau_1 - 1)) \times \cdots \times (I_N + 2(\tau_N - 1))}$ is a $N$-dimensional reflection padding operator[1] of tensor, $\text{unfold}_{(D,T)} : \mathbb{R}^{\tau_1(I_1 + \tau_1 - 1) \times \cdots \times \tau_N(I_N + \tau_N - 1)} \to \mathbb{R}^{D \times T}$ is an unfolding operator which outputs a matrix from an input $N$-th order tensor, and $\boldsymbol{S}_n \in \mathbb{R}^{\tau_n(I_n + \tau_n - 1) \times (I_n + 2(\tau_n - 1))}$ is a duplication matrix. Fig. 10 shows the duplication matrix with $\tau$.

For example, our Hankelization with reflection padding of $\boldsymbol{f} = [f_1, f_2, ..., f_7]$ with $\tau = 3$ is given by

$$
\begin{pmatrix}
f_3 & f_2 & f_1 & f_2 & f_3 & f_4 & f_5 & f_6 & f_7 \\
f_2 & f_1 & f_2 & f_3 & f_4 & f_5 & f_6 & f_7 & f_6 \\
f_1 & f_2 & f_3 & f_4 & f_5 & f_6 & f_7 & f_6 & f_5
\end{pmatrix}.
\tag{11}
$$

Fig. 11 shows an example of our multiway-delay embedding in case of second order tensors. The overlapped patch grid is constructed by multi-linear tensor product with $\boldsymbol{S}_n$. Finally, all patches are splitted, lined up, and vectorized.

The Moore-Penrose pseudo inverse of $\mathcal{H}$ is given by

$$
\mathcal{H}^{\dagger}(\boldsymbol{H}) = \text{trim}_{\boldsymbol{\tau}}(\text{fold}_{(D,T)}(\boldsymbol{H}) \times_1 \boldsymbol{S}_1^{\dagger} \cdots \times_N \boldsymbol{S}_N^{\dagger}),
\tag{12}
$$

where $\boldsymbol{S}_n^{\dagger} := (\boldsymbol{S}_n^T \boldsymbol{S}_n)^{-1} \boldsymbol{S}_n^T$ is a pseudo inverse of $\boldsymbol{S}_n$, $\text{fold}_{(D,T)} := \text{unfold}_{(D,T)}^{-1}$, and $\text{trim}_{\boldsymbol{\tau}} = \text{pad}_{\boldsymbol{\tau}}^{\dagger}$ is a trimming operator for removing $(\tau_n - 1)$ elements at start and end of each mode. Note that $\mathcal{H}^{\dagger} \circ \mathcal{H}$ is an identity map, but $\mathcal{H} \circ \mathcal{H}^{\dagger}$ is not, that is kind of a projection.

### B.1 Delay embedding using convolution

Delay embedding and its pseudo inverse can be implemented by using convolution with all one-hot-tensor windows of size $(\tau_1, \tau_2, ..., \tau_N)$. The one-hot-tensor windows can be given by folding a $D$-dimensional identity matrix $\boldsymbol{I}_D \in \mathbb{R}^{D \times D}$ into $\boldsymbol{\mathcal{I}}_D \in \mathbb{R}^{\tau_1 \times \cdots \times \tau_N \times D}$. Fig. 12 shows a calculation flow of multi-way delay embedding using convolution in a case of $N = 2$. Multi-linear tensor product is replaced with convolution with one-hot-tensor windows.

---

[1]For one dimensional array $\boldsymbol{x} = [x_1, ..., x_I]^T$, we have $\text{pad}_{\tau}(\boldsymbol{x}) = [\underbrace{x_{\tau}, ..., x_2}_{\tau-1}, \underbrace{x_1, ..., x_I}_{I}, \underbrace{x_{I-1}, ..., x_{I-\tau}}_{\tau-1}]^T$.

Pseudo inverse of the convolution with padding is given by its adjoint operation, which is called as the "transposed convolution" in some neural network library, with trimming and simple scaling with $D^{-1}$.

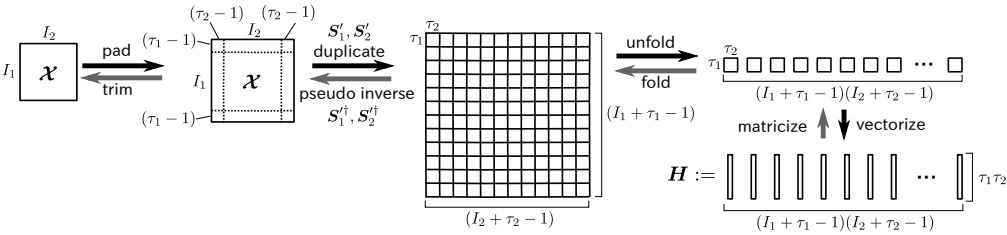

Figure 10: Duplication matrix. In case that we have $I$ columns, it consists of $(I - \tau + 1)$ identity matrices of size $(\tau, \tau)$.

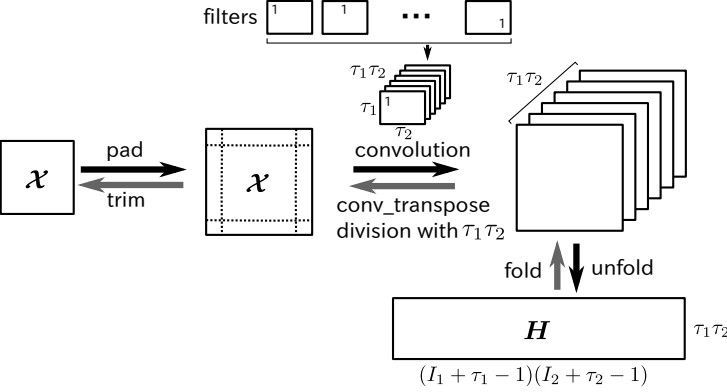

Figure 11: Flow of multiway-delay-embedding operation ($N = 2$).

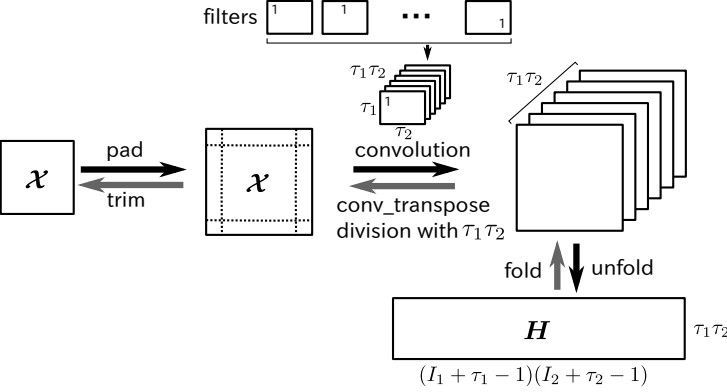

Figure 12: Multiway-delay-embedding using convolution ($N = 2$).

## C  DESIGN OF AUTO-ENCODER

In this section, we discuss how to design the neural network architecture of auto-encoder for restricting the manifold $\mathcal{M}_r$. The simplest way is controlling the value of $r$, and it directly restricts the dimensionality of latent space. There are many other possibilities: Tikhonov regularization (Goodfellow et al., 2016), drop-out (Gal & Ghahramani, 2016), denoising auto-encoder (Vincent et al., 2008), variational auto-encoder (Diederik P Kingma, 2014), adversarial auto-encoder (Makhzani et al., 2015), alpha-GAN (Rosca et al., 2017), and so on. All methods have some perspective and promise, however the cost is not low. In this study, we select an attractive and fundamental one: "denoising auto-encoder"(DAE) (Vincent et al., 2008). The DAE is attractive because it has a strong relationship with Tikhonov regularization (Bishop, 1995), and decreases the entropy of data (Sonoda & Murata, 2017). Furthermore, learning with noise is also employed in the deep image prior.

Finally, we designed an auto-encoder with controlling the dimension $r$ and the standard deviation $\sigma$ of additive zero-mean Gaussian noise. Fig. 13 shows the illustration of an example of architecture of

auto-encoder which we used in this study. In this case, it consists of five hidden variables of which sizes are $[D, D, r, D, D]$ with leaky ReLU activation.

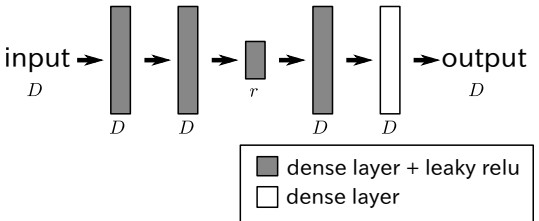

Figure 13: An example of architecture of auto-encoder.

## D  A SPECIAL SETTING FOR COLOR-IMAGE RECOVERY

In case of multi-channel or color image recovery case, we use a special setting of generator network because spacial pattern of individual channels are similar and the patch-manifold can be shared. Fig. 14 shows an illustration of the auto-encoder shared version of MMES in a case of color image recovery. In this case, we put three channels of input and each channel input is embedded, independently. Then, three block Hankel matrices are concatenated, and auto-encoded simultaneously. Inverted three images are stacked as a color-image (third-order tensor), and finally color-transformed. The last color-transform can be implemented by convolution layer with kernel size (1,1), and it is also optimized as parameters. It should be noted that the input three channels are not necessary to correspond to RGB, but it would be optimized as some compact color-representation.

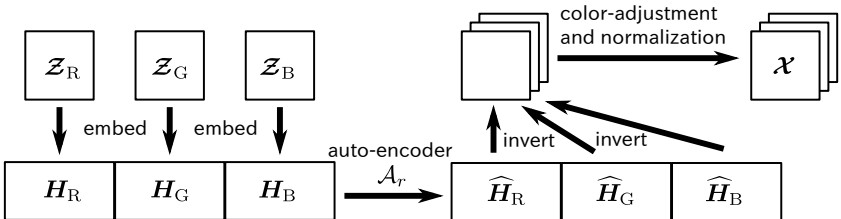

Figure 14: Generator network in a case of color-image recovery.

## E  OTHER DETAILS OF IMAGE-INPAINTING EXPERIMENTS

Here, we explain detailed experimental settings in Section 4.2.

In this section, we compared performance of the proposed method with several selected unsupervised image inpainting methods: low-rank tensor completion (HaLRTC) (Liu et al., 2013), parallel low-rank matrix factorization (TMac) (Xu et al., 2015), tubal nuclear norm regularization (tSVD) (Zhang et al., 2014b), Tucker decomposition with rank increment (Tucker inc.) (Yokota et al., 2018), low-rank and total-variation (LRTV) regularization[2] (Yokota & Hontani, 2017; 2019), smooth PARAFAC tensor completion (SPC)[3] (Yokota et al., 2016), GSR[4] (Zhang et al., 2014a), multi-way

---

[2]For LRTV, the MATLAB software was downloaded from `https://sites.google.com/site/yokotatsuya/home/software/lrtv_pds`

[3]For SPC, the MATLAB software was downloaded from `https://sites.google.com/site/yokotatsuya/home/software/smooth-parafac-decomposition-for-tensor-completion`.

[4]For GSR, each color channel was recovered, independently, using the MATLAB software downloaded from `https://github.com/jianzhangcs/GSR`.

Table 2: Parameter settings for MMES in image completion experiments

| $(\tau,r)$ | airplane | baboon | barbara | facade | house | lena | peppers | saiboat |
|---|---|---|---|---|---|---|---|---|
| 50 % | (16,4) | (10,4) | (6,4) | (10,4) | (16,4) | (6,4) | (6,4) | (6,4) |
| 70 % | (16,4) | (10,4) | (6,4) | (16,4) | (16,4) | (6,4) | (16,4) | (6,4) |
| 90 % | (16,4) | (4,8) | (6,4) | (16,4) | (16,4) | (8,4) | (16,4) | (4,4) |
| 95 % | (16,4) | (4,6) | (6,4) | (16,4) | (16,4) | (6,8) | (16,4) | (6,8) |
| 99 % | (8,32) | (4,4) | (6,4) | (4,1) | (8,16) | (10,32) | (8,8) | (6,4) |

delay embedding based Tucker modeling (MDT-Tucker)[5] (Yokota et al., 2018), and DIP[6] (Ulyanov et al., 2018).

For this experiments, hyper-parameters of all methods were tuned manually to perform the best peak-signal-to-noise ratio (PSNR) and for structural similarity (SSIM), although it would not be perfect. For DIP, we did not try the all network structures with various kernel sizes, filter sizes, and depth. We just employed "default architecture", which the details are available in supplemental material[7] of (Ulyanov et al., 2018), and employed the best results at the appropriate intermediate iterations in optimizations based on the value of PSNR. For the proposed MMES method, we adaptively selected the patch-size $\tau$, and dimension $r$. Table 2 shows parameter settings of $\boldsymbol{\tau} = [\tau, \tau]$ and $r$ for MMES. Noise level of denoising auto-encoder was set as $\sigma = 0.05$ for all images. For auto-encoder, same architecture shown in Fig. 13 was employed. Initial learning rate of Adam optimizer was 0.01 and we decayed the learning rate with 0.98 every 100 iterations. The optimization was stopped after 20,000 iterations for each image.

## F    OTHER DETAILS OF SUPER-RESOLUTION EXPERIMENTS

Here, we explain detailed experimental settings in Section 4.3.

In this section, we compare performance of the proposed method with several selected unsupervised image super-resolution methods: bicubic interpolation, GSR[8] (Zhang et al., 2014a), ZSSR[9] and DIP (Ulyanov et al., 2018).

In this experiments, DIP was conducted with the best number of iterations from $\{1000, 2000, 3000, ..., 9000\}$. For four times (x4) up-scaling in MMES, we set $\tau = 6$, $r = 32$, and $\sigma = 0.1$. For eight times (x8) up-scaling in MMES, we set $\tau = 6$, $r = 16$, and $\sigma = 0.1$. For all images in MMES, the architecture of auto-encoder consists of three hidden layers with sizes of $[8\tau^2, r, 8\tau^2]$. We assumed the same Lanczos2 kernel for down-sampling system for all super-resolution methods.

Tab. 3 shows values of PSNR and SSIM of the results. We used three (256,256,3) color images, and six (512,512,3) color images. Super resolution methods scaling up them from four or eight times down-scaled images of them. According to this quantitative evaluation, bicubic interpolation was clearly worse than others. ZSSR was good for (128,128,3) color images, however the performance were substantially decreased for (64,64,3) color image. Basically, GSR, DIP, and MMES were very competitive. In detail, DIP was slightly better than GSR, and the proposed MMES was slightly better than DIP.

---

[5]For MDT-Tucker, the MATLAB software was downloaded from `https://sites.google.com/site/yokotatsuya/home/software/mdt-tucker-decomposition-for-tensor-completion`.

[6]For DIP, we implemented by ourselves in Python with `TensorFlow`.

[7]`https://dmitryulyanov.github.io/deep_image_prior`

[8]For GSR, each color channel was recovered, independently, using the MATLAB software downloaded from `https://github.com/jianzhangcs/GSR`. We slightly modified its MATLAB code for applying it to super-resolution task.

[9]For ZSSR, software was downloaded from `https://github.com/assafshocher/ZSSR`. We set the same Lanczos2 kernel for this super-resolution task.

Table 3: Values of PSNR and SSIM in super-resolution task

| PSNR / SSIM | Bicubic | GSR | ZSSR | DIP | MMES (proposed) |
|---|---|---|---|---|---|
| Starfish (64 to 256) | 23.98 / .7124 | 25.73 / .7922 | 25.13 / .7748 | 25.79 / .7930 | **26.18 / .8099** |
| House (64 to 256) | 26.21 / .7839 | 28.05 / .8394 | 26.89 / .8202 | 28.33 / .8420 | **28.79 / .8448** |
| Leaves (64 to 256) | 19.10 / .6673 | 22.60 / .8511 | 20.19 / .7406 | 22.54 / .8535 | **23.96 / .8935** |
| Airplane (128 to 512) | 26.30 / .9176 | 27.74 / .9487 | 27.53 / .9430 | 27.49 / .9375 | **28.40 / .9503** |
| Airplane (64 to 512) | 22.93 / .7545 | 23.79 / .8061 | 22.74 / .7629 | 23.83 / .8155 | **24.10 / .8207** |
| Baboon (128 to 512) | 20.61 / .6904 | 20.93 / **.7542** | 20.94 / .7489 | 20.52 / .7260 | 20.92 / .7486 |
| Baboon (64 to 512) | 19.38 / .4505 | 19.61 / .5039 | 19.54 / .4926 | **19.64 / .5085** | **19.64** / .5024 |
| Lena (128 to 512) | 28.64 / .9172 | 30.36 / **.9481** | 29.56 / .9417 | 29.91 / .9406 | 29.76 / .9406 |
| Lena (64 to 512) | 25.23 / .7710 | 26.47 / .8271 | 25.56 / .7946 | **26.71 / .8340** | 26.68 / .8327 |
| Monarch (128 to 512) | 24.88 / .9322 | 27.67 / .9679 | 26.00 / .9514 | 27.90 / .9576 | **28.81 / .9686** |
| Monarch (64 to 512) | 20.65 / .7697 | 22.13 / .8393 | 21.22 / .8018 | 22.65 / .8594 | **23.01 / .8627** |
| Peppers (128 to 512) | 27.27 / .9392 | **29.19 / .9642** | 28.60 / .9589 | 28.78 / .9578 | 28.85 / .9584 |
| Peppers (64 to 512) | 24.15 / .8173 | 25.52 / .8753 | 24.35 / .8299 | **26.07 / .8904** | 25.75 / .8794 |
| Sailboat (128 to 512) | 24.38 / .8885 | 25.43 / .9262 | 25.33 / .9228 | 25.13 / .9130 | **25.72 / .9273** |
| Sailboat (64 to 512) | 21.22 / .6898 | 21.94 / .7463 | 21.55 / .7276 | 22.32 / .7664 | **23.37 / .7705** |
| Average | 23.66 / .7801 | 25.14 / .8393 | 24.35 / .8141 | 25.19 / .8401 | **25.53 / .8474** |

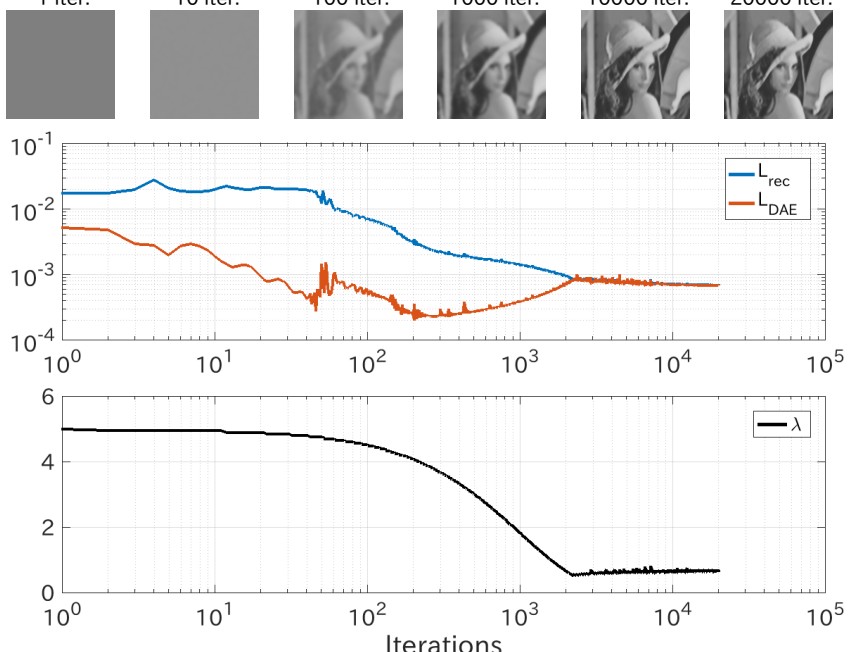

Figure 15: Optimization behavior.

# G OTHER EXPERIMENTAL RESULTS

## G.1 OPTIMIZATION BEHAVIOR

For this experiment, we recovered 50% missing gray-scale image of 'Lena'. We stopped the optimization algorithm after 20,000 iterations. Learning rate was set as 0.01, and we decayed the learning rate with 0.98 every 100 iterations. $\lambda$ was adapted by Algorithm 1 every 10 iterations. Fig. 15 shows optimization behaviors of reconstructed image, reconstruction loss $\mathcal{L}_{\text{rec}}$, auto-encoding loss $\mathcal{L}_{\text{DAE}}$, and trade-off coefficient $\lambda$. By using trade-off adjustment, the reconstruction loss and the auto-encoding loss were intersected around 1,500 iterations, and both losses were jointly decreased after the intersection point.

## G.2 HYPER-PARAMETER SENSITIVITY

We evaluate the sensitivity of MMES with three hyper-parameters: $r$, $\sigma$, and $\tau$. First, we fixed the patch-size as $(8, 8)$, and dimension $r$ and noise standard deviation $\sigma$ were varied. Fig. 17 shows

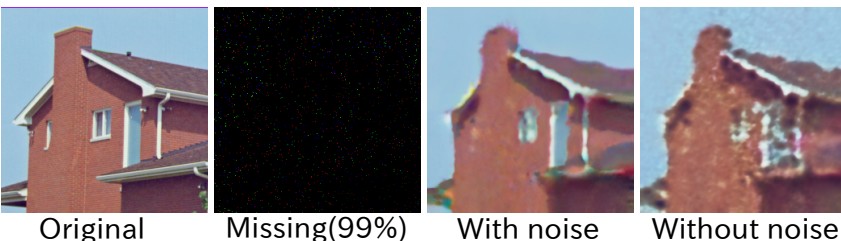

Original  Missing(99%)  With noise  Without noise

Figure 16: Reconstruction of 'home' image by training with/without noise in deep image prior.

the reconstruction results of a 99% missing image of 'Lena' by the proposed method with different settings of $(r, \sigma)$. The proposed method with very low dimension ($r = 1$) provided blurred results, and the proposed method with very high dimension ($r = 64$) provided results which have many peaks. Furthermore, some appropriate noise level ($\sigma = 0.05$) provides sharp and clean results. For reference, Fig. 16 shows the difference of DIP optimized with and without noise. From both results, the effects of learning with noise can be confirmed.

Next, we fixed the noise level as $\sigma = 0.05$, and the patch-size were varied with some values of $r$. Fig. 18 shows the results with various patch-size settings for recovering a 99% missing image. The patch sizes $\tau$ of (8,8) or (10,10) were appropriate for this case. Patch size is very important because it depends on the variety of patch patterns. If patch size is too large, then patch variations might expand and the structure of patch-manifold is complicated. By contrast, if patch size is too small, then the information obtained from the embedded matrix $\boldsymbol{H}$ is limited and the reconstruction becomes difficult in highly missing cases. The same problem might be occurred in all patch-based image reconstruction methods (Buades et al., 2005; Dabov et al., 2007; Gu et al., 2014; Zhang et al., 2014a). However, good patch sizes would be different for different images and types/levels of corruption, and the estimation of good patch size is an open problem. Multi-scale approach (Yair & Michaeli, 2018) may reduce a part of this issue but the patch-size is still fixed or tuned as a hyper-parameter.

### G.3    VOLUMETRIC/3D IMAGE/TENSOR COMPLETION

In this section, we show the results of MR-image/3D-tensor completion problem. The size of MR image is (109,91,91). We randomly remove 50%, 70%, and 90% voxels of the original MR-image and recover the missing MR-images by the proposed method and DIP. For DIP, we implemented the 3D version of *default architecture* in `TensorFlow`, but the number of filters of shallow layers were slightly reduced because of the GPU memory constraint. For the proposed method, 3D patch-size was set as $\tau = [4, 4, 4]$, the lowest dimension was $r = 6$, and noise level was $\sigma = 0.05$. Same architecture shown in Fig. 13 was employed.

Fig. 19 shows reconstruction behavior of PSNR with final value of PSNR/SSIM in this experiment. From the values of PSNR and SSIM, the proposed MMES outperformed DIP in low-rate missing cases, and it is quite competitive in highly missing cases. The some degradation of DIP might be occurred by the insufficiency of filter sizes since much more filter sizes would be required for 3D ConvNet than 2D ConvNet. Moreover, computational times required for our MMES were significantly shorter than that of DIP in this tensor completion problem.

noise levels of denoising autoencoder

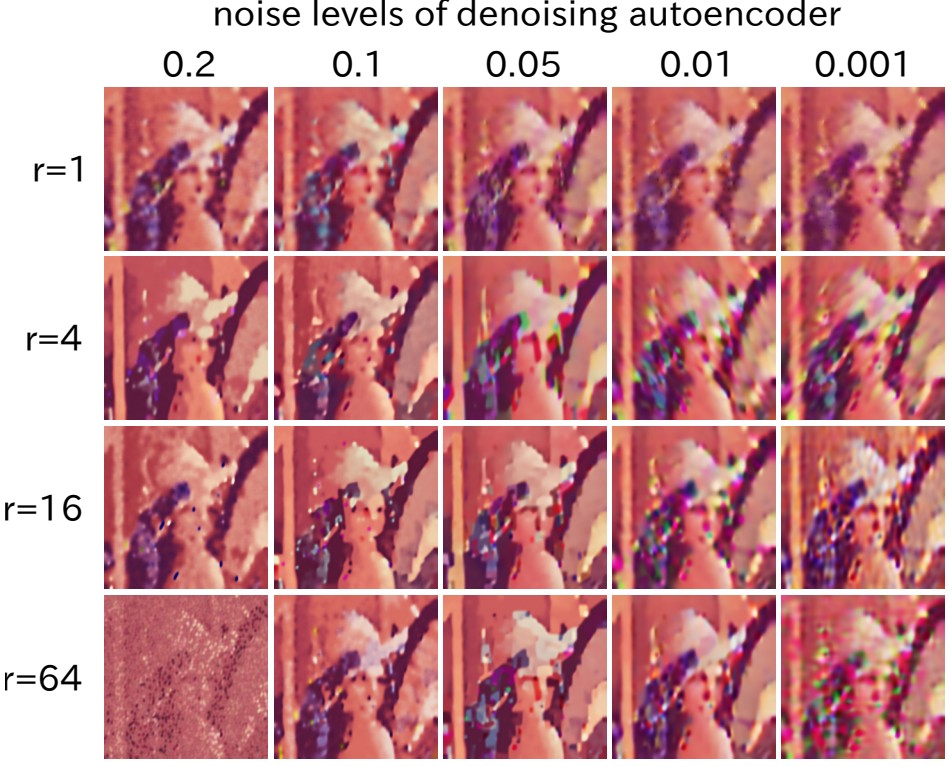

Figure 17: Performance of reconstruction of color image of 'Lena' with 99% pixels missing for various parameter setting.

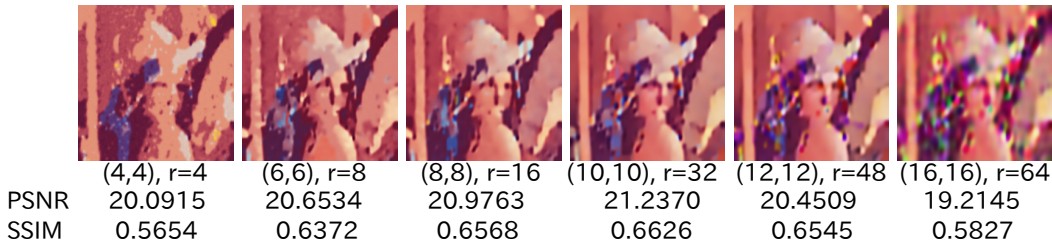

Figure 18: Reconstruction of 'Lena' image for various patch sizes $\tau$.

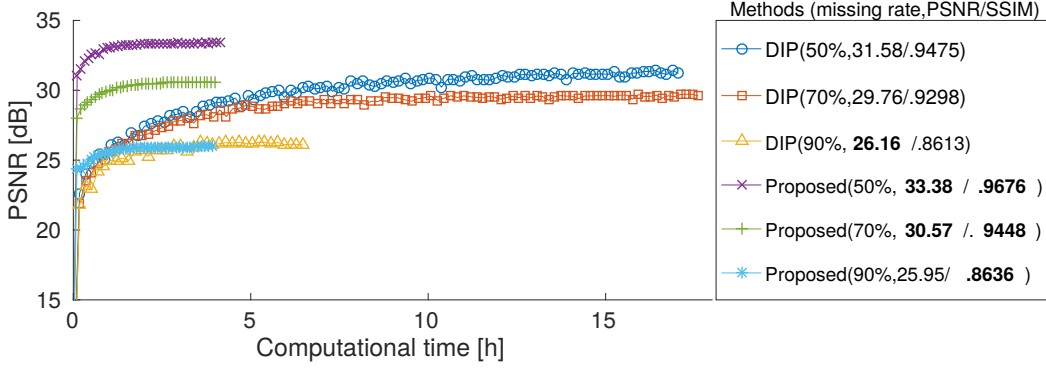

Figure 19: Results of MRI completion: Optimization behaviors of PSNR with final values of PSNR/SSIM by DIP and proposed MMES.

