# OpenReview forum: "Manifold Modeling in Embedded Space: A Perspective for Interpreting "Deep Image Prior""
_ICLR.cc/2020/Conference — Reject_

### Official Review · AnonReviewer2 · 2019-10-23
**Official Blind Review #2**

**Rating:** 6

**Review:**


This paper proposed a low-dimensional patch-manifold prior perspective for reinterpreting the deep image prior. I think this is very interesting work that could have a lot of impact in vision and beyond, since effective construct the problem in reconstruction tasks are highly relevant to a number of tasks. I was initially quite excited about this paper, but as I drilled into the details of this work.

For the manifold modeling, though the authors defined each part of the formulation in equations (1)-(5), but it is not clear how to design corresponding efficient structures for different low-level problems. In other words, the application ability is not clear. For example, if the proposed MMES scheme is used in image deconvolution tasks, how to design the corresponding structure?

For the parameters, there is no description of the parameters set in subsection 4.2 and 4.3. And what are the criteria for selecting these parameters?

For the experimental part, the comparison experiments in subsection 4.1(Toy examples) and subsection 4.3 (Color image superresolution) lack comparisons with the latest methods. I think, more comparison experiments should be provided.


[Update after rebuttal period]
The revision and responses have addressed most of my concerns, so I keep my original score.




**Experience Assessment:**

I have published in this field for several years.

**Review Assessment: Checking Correctness Of Derivations And Theory:**

I assessed the sensibility of the derivations and theory.

**Review Assessment: Checking Correctness Of Experiments:**

I assessed the sensibility of the experiments.

**Review Assessment: Thoroughness In Paper Reading:**

I read the paper at least twice and used my best judgement in assessing the paper.

---

> ### Author Response · Authors · 2019-11-08
> **Response to summary**
>
> Thank you very much for encouraging us comments. We agree that  that there are some  issues which need to be explained  with more detail or more precisely.

---

> ### Author Response · Authors · 2019-11-08
> **About application ability**
>
> Thank you for this insightful comment.  We agree with you that the application ability is very important. In our experiments we considered only inpainting, denoising and super-resolution. Image deconvolution so far was out of the scope of this paper, however super-resolution is a special case of image deconvolution in case of blur kernel is Lanczos kernel with sub-sampling.  The good structure designs for deconvolution and super-resolution would be similar.  We can conduct image deconvolution experiment later, but we are not sure it can be done within two weeks.

---

> ### Author Response · Authors · 2019-11-08
> **About hyper parameters**
>
> In general, the proposed method is not very sensitive to set of parameters and work very well for wide variability of set parameters. This was confirmed by our extensive computer simulation experiments.  However, to chose optimal set of parameters like \tau is still open challenging problem. In general, the key factor is to make appropriate "pre-processing: via block tensor Hankelization, which allows us to exploit local correlations between various patches and to chose parameters or internal convolutional AE.

---

> ### Author Response · Authors · 2019-11-08
> **Additional comparison**
>
> We agree with you for adding comparison in super-resolution. We can provide more experiments later, we are not able in two week to perform more quite sophisticated experiments. We are quite convinced that proposed method works consistently well for wide spectrum of problems and data sets.
>
> We plan to add zero-shot super resolution (ZSSR).
> -- Assaf Shocher, Nadav Cohen, and Michal Irani. Zero-shot super-resolution using deep internal learning. In Proceedings of CVPR, pp. 3118–3126, 2018.

---

### Official Review · AnonReviewer3 · 2019-10-24
**Official Blind Review #3**

**Rating:** 6

**Review:**

In this paper, the authors present a natural image model based on the manifold of image patches.  It is similar to the Deep Image Prior in that it is untrained and has a convolutional-like structure.  It leads to an optimization problem with a reconstruction loss term and an auto encoding term.  The authors show empirical results in time series recovery, non-semantic inpainting, and super resolution.  In the image processing tasks, the performance of the proposed algorithm is on par (sometimes slightly worse, sometimes slightly better) than that of DIP.

I think the perspective of image patch analysis is a useful addition to the knowledge base for unlearned image priors.  That said, the paper says it tackles the questions for why the DIP "works so well" and why convolution operations are "essential for image reconstruction or enhancement".  After reading the paper, it is unclear to me how this work addresses these questions.  In particular, demonstrating a similar convolutional system does not rule out the possibility that there are non-convolutional systems that also explain the effect.  For example, the Deep Geometric Prior paper is nonconvolutional (it is entirely a MLP), and it also has the effect of fitting a smooth signal without training (subject to early stopping).  The DGP could be applied to images as well, resulting in a nonconvolutional deep image prior.  I think the authors should address more clearly and thoroughly the logical connection between their results and the explanation of the DIP, especially in light of the DGP.

The paper claims that the proposed method is more interpretable, and it would be nice if they could demonstrate this interpretability and the benefits it brings in solving image reconstruction problems.

As a result, I am inclined to recommend a weak reject, but if the concerns above are addressed, I envision my rating could improve upon rebuttal.

**Experience Assessment:**

I have published one or two papers in this area.

**Review Assessment: Checking Correctness Of Derivations And Theory:**

I assessed the sensibility of the derivations and theory.

**Review Assessment: Checking Correctness Of Experiments:**

I assessed the sensibility of the experiments.

**Review Assessment: Thoroughness In Paper Reading:**

I read the paper at least twice and used my best judgement in assessing the paper.

---

> ### Author Response · Authors · 2019-11-08
> **Reponse to summary**
>
> Thank you for very insightful and useful comments.
> We would like to mention that the main objective of this paper was not to dramatically improve the performance but rather provide new insight and explanation of DIP model and also propose possible extensions or modifications.

---

> ### Author Response · Authors · 2019-11-08
> **Discussion about convolutional and non-convolutional networks**
>
> Thank you for the insightful comments. Our MMES has many connection with DIP model such as embedding, non-linear mapping and the training with noise. In fact, we show that our Hankelization operations are equivalent to 1D and 2D convolution. Extension of such transformation (from convolution to block Hankel tensors) for 3D or even N-D convolutions is principally possible. So, in general convolution can be replaced by other operations approximately or precisely and even other operations are possible, but this need further detailed research.
>
> We partially agree with the comments.  Essences of DIP model would not be only a convolution but also a deep network.  The deep geometric prior (DGP) [a1] utilises a good properties of deep network (MLP) for shape reconstruction which efficiently learn a smooth function from 2D space to 3D space.  For example, it can be used for gray scale image reconstruction if an image is regarded as point could in 3D space (x,y,intensity).  However, this may not provide the good image reconstruction like DIP, because it just smoothly interpolates a point cloud by surface like a Volonoi interpolation (you can check the difference between Voronoi image inpainting and DIP inpainting at https://dmitryulyanov.github.io/deep_image_prior ).  Especially it can not provide a property of "self-similarity" in natural image.
>
> Another example would be deep decoder [a2].  Although a "non-convolutional" network architecture with up-sampling layers is considered for image reconstruction, Authors emphasize the closed relationship between convolutional layer and upsampling layer.  Authors said "If there is no upsampling layer, then there is no notion of locality in the resultant image" in deep decoder. It implies the "locality" is the essence of image model, and the convolution/upsampling layer provides it. Furthermore, the deep decoder has a close relationship with our MMES. Note that the MMES is originally/essentially has only decoder and inverse MDT (see eq.(3)), and the encoder is just used for satisfying Hankel structure.  The decoder and inverse MDT in our MMES are respectively corresponding linear operation and upsampling layer in deep decoder.  Both concepts are very similar.
>
> From this, we can say the essence of image model is the "locality". And its locality can be provided by "convolution" or "delay-embedding".
>
> Thus, convolutional architecture is still essential.  In ICCV2019, two generative models: SinGAN [a3] and InGAN [a4] learned from only a single image, have been proposed.  Key concept of both papers is to impose the constraint for local patches of image to be natural.  From a perspective of the constraint for local patches of image, our MMES has closed relationship with these works. However, we explicitly impose a low-dimensional manifold constraint for local patches rather than adversarial training with patch discriminators.
>
> We will add above discussions in revised version with references.
>
> [a1] Williams, Francis, et al. "Deep geometric prior for surface reconstruction." Proceedings of the IEEE Conference on Computer Vision and Pattern Recognition. 2019.
> [a2] Heckel, Reinhard, and Paul Hand. "Deep decoder: Concise image representations from untrained non-convolutional networks." arXiv preprint arXiv:1810.03982 (2018), published in ICLR2019.
> [a3] Shaham, Tamar Rott, Tali Dekel, and Tomer Michaeli. "SinGAN: Learning a Generative Model from a Single Natural Image." Proceedings of the IEEE International Conference on Computer Vision. 2019.
> [a4] Shocher, Assaf, et al. "InGAN: Capturing and Retargeting the DNA of a Natural Image." Proceedings of the IEEE International Conference on Computer Vision. 2019.

---

> ### Author Response · Authors · 2019-11-08
> **About interpretability**
>
> We agree that this interpretability should be demonstrated more clearly and more explicitly.
> We attempted to interpret the method mathematically by formulating constrained optimization problem and provided algorithm to solve this problem and proposed associated deep neural network (AE) model. For these purposes, we exploited the concept of manifold modeling and investigated similarity between patches located on this manifold.
>
> We hope that the proposed architecture of a new AE is quite promising for many potential applications not only for image reconstructions but also times series/video reconstruction and time series/video forecasting.

---

### Official Review · AnonReviewer4 · 2019-11-12
**Official Blind Review #4**

**Rating:** 6

**Review:**

This paper introduces a transformation from the deep image prior (DIP) to an embedding with an autoencoder (MMES). The authors aim to use this transformation to explain ("in words") why the DIP works so well and explain why convolutions are needed in the DIP. The contributions are summarised as a) providing an interpretable analogue to the convnet, b) demonstration of the proposed method's effectiveness, and c) characterisation of the DIP as a "low-dimensional patch-manifold prior".

I think the MMES approach is interesting and potentially a good analogue to the DIP, and explicitly draws out the locality prior the authors claim is integral to DIP. The good results and comparison to DIP demonstrates that this locality prior may be important to the task.

I disagree that this method is "interpretable"/"explainable", at least without any evidence toward this presented in the paper. There is still fundamentally a deep network as in DIP. The discussion on interpretability is limited and mostly provided through comparison with other methods.

All up I think this is a useful paper, even though the paper overstates its contributions. I would like to see the clarity improved: it took me a long while to make the connection between the method presented in the paper and the implications for DIP. This connection should have been more explicit in the paper. I would also like to see the "interpretability" statement either clearly explained or removed (I am not convinced that this method is interpretable).

**Experience Assessment:**

I do not know much about this area.

**Review Assessment: Checking Correctness Of Derivations And Theory:**

I assessed the sensibility of the derivations and theory.

**Review Assessment: Checking Correctness Of Experiments:**

I assessed the sensibility of the experiments.

**Review Assessment: Thoroughness In Paper Reading:**

I read the paper at least twice and used my best judgement in assessing the paper.

---

> ### Author Response · Authors · 2019-11-13
> **Our contributions about interpretability**
>
> Thank you for your insightful and encouraging comments.  We are very happy that you recognized our contributions in this study related to the proposed MMES approach.
>
> The main concern is the interpretability of DIP and our related model.  We agree with you that our interpretation or explanation of DIP is quite limited in this study. The links between DIP and MMES were shown by extensive computer experiments (the both model can perform the same tasks like denoising, inpainting, reconstruction, superresolution and deconvolutions), and also mathematical connections, especially equivalent transformations between convolution operations and Hankelization. Our contribution is to show some new insight of DIP and/or to indicate that there are some "perspectives" to interpret DIP (at least partially) through the manifold modeling. In other words, we believe that manifold modeling and learning may lead to partial interpretations of some deep learnng models.
>
> It is well known that there is no mathematical definition of interpretability in machine learning and there is no one unique definition of interpretation. We understand by the interpretability a degree to which a human can consistently predict the model’s results or performance. The higher the interpretability of a deep learning model, the easier it is for someone to comprehend why certain performance or predictions or expected output can be achieved. We think that a model is better interpretable than another model if its performance or behaviors are easier for a human to comprehend than performance of the other models.
>
> We will revise our manuscript to explain the contributions more precisely, and strengthen the explanations/characterizations about MMES in more details.
>
> A discussion about the our (partial) interpretation of MMES is as follow:
> 1) From a perspective of dememsionality reduction/manifold learning
> The manifold learning and associated auto-encoder (AE) can be viewed as the generalized non-linear version of principal component analysis (PCA). In fact, manifold learning solves the key problem of dimensionality reduction very efficiently. In other words, manifold learning (modeling) is an approach to non-linear dimensionality reduction. Manifold modeling for this task are based on the idea that the dimensionality of many data sets is only artificially high. Although the patches of images (data points) consist of handreds/thousands pixels, they may be represented as a function of only a few or quite limited number underlying parameters. That is, the patches are actually samples from a low-dimensional manifold that is embedded in a high-dimensional space. Manifold learning algorithms attempt to uncover these parameters in order to find a low dimensional representation of the images.
>
> In our approach to solve the problem we applied original embedding via multi-way delay embedding transform (MDT or Hankelization). Our algorithm is based on the optimization of costs (loss) function and it works towards extracting the low-dimensional manifold that is used to describe the high-dimensional data. The manifold is described mathematically by Eq.(2) and loss (objective) function is formulated by Eqs (1) - (5).
>
> In other words, manifold learning can be thought of as a natural generalization of linear frameworks like PCA to fit to non-linear structure in data. Though supervised variants exist, our manifold learning problem is unsupervised: It learns the high-dimensional structure of the data from the available data itself, without the use of predetermined classifications.

---

> ### Author Response · Authors · 2019-11-13
> **Our contributions about interpretability (cont'd)**
>
> 2) Regarding our attempt to interpret "noise impedance in DIP" via MMES:
> As mentioned at introduction of our manuscript, DIP paper reports an important phenomenon of noise impedance of ConvNet structures. Let us consider the sparse-land model, i.e. noise-free images are distributed along low-dimensional manifolds in the high-dimensional Euclidean space and images perturbed by noises thicken the manifolds (make the dimension of the manifolds higher). Under this model, the distribution of images can be assumed to be higher along the low-dimensional noise-free image manifolds. When we assume that the image patches are sampled from low-dimensional manifold like sparse-land model, it is difficult to put noisy patches on the low-dimensional manifold. Let us consider to fit the network for noisy images. In such case the fastest way for decreasing squared error (loss function) is to learn "similar patches" which often appear in a large set of image-patches. Note that finding similar image-patches for denoising is well-known problem solved, e.g., by BM3D algorithm, which find similar image patches by template matching. In contrast, our auto-encoder automatically maps similar-patches into close points on the low-dimensional manifold. When similar-patches have some noise, the low-dimensional representation tries to keep the common components of similar patches, while reducing the noise components. This has been proved by Alein and Bengio so that a (denoising) autoencoder maps input image patches toward higher density portions in the image space. In other words, a (denoising) auto-encoder has kind of a force to reconstruct the low-dimensional patch manifold, and this is our rough explanation of noise impedance phenomenon. Although the proposed MMES and DIP are not completely equivalent, we see many analogies and similarities and we believe that our MMES model and associated learning algorithm give some new insight for DIP.

---

### Author Response · Authors · 2019-11-14
**Revised manuscript has been uploaded, and its summary of changes**

We are really appreciate with Reviewers and Aria chair for their effort to evaluate our study.
Here, we will summarise main updates of revised manuscript in accordance to the Reviewer's comments.

 - We added a state-of-the-art super-resolution method named ZSSR [b1] for comparison in accordance to the concern from Reviewer #2.
 - We added color-image deconvolution experiments comparing DIP and MMES in accordance to the concern from Reviewer #2.
 - We improved the related works, and discussion sections in accordance to concern from Reviewer #3.
 - We improved the introduction, discussion sections in accordance to concern from Reviewer #4.
 - We carefully checked about the statement of interpretation in accordance to concern from Reviewer #4.
 - We added a new section for discussing the interpretation of MMES and connection to DIP in accordance to concern from Reviewer #4.

We are happy if you are convinced with this revisions.

[b1] Shocher, Assaf, Nadav Cohen, and Michal Irani. "“zero-shot” super-resolution using deep internal learning." Proceedings of the IEEE Conference on Computer Vision and Pattern Recognition. 2018.

---

### Decision · Program_Chairs · 2019-12-19

**Decision:**

Reject

**Comment:**

The paper proposes a combination of a delay embedding as well as an autoencoder to perform representation learning. The proposed algorithm shows competitive performance with deep image prior, which is a convnet structure. The paper claims that the new approach is interpretable and provides explainable insight into image priors.

The discussion period was used constructively, with the authors addressing reviewer comments, and the reviewers acknowledging this an updating their scores.

Overall, the proposed architecture is good, but the structure and presentation of the paper is still not up to the standards of ICLR. The current presentation seems to over-claim interpretability, without sufficient theoretical or empirical evidence.